# Infection, Inflammation, and Immunity in Sepsis

**DOI:** 10.3390/biom13091332

**Published:** 2023-08-31

**Authors:** Undurti N. Das

**Affiliations:** UND Life Sciences, 2221 NW 5th St., Battle Ground, WA 98604, USA; undurti@hotmail.com or undurti@lipidworld.com; Tel.: +1-508-904-5376

**Keywords:** sepsis, radiation, cytokines, eicosanoids, lipoxins, resolvins, cGAS-STING system, inflammation, wound healing

## Abstract

Sepsis is triggered by microbial infection, injury, or even major surgery. Both innate and adaptive immune systems are involved in its pathogenesis. Cytoplasmic presence of DNA or RNA of the invading organisms or damaged nuclear material (in the form of micronucleus in the cytoplasm) in the host cell need to be eliminated by various nucleases; failure to do so leads to the triggering of inflammation by the cellular cGAS-STING system, which induces the release of IL-6, TNF-α, and IFNs. These cytokines activate phospholipase A2 (PLA2), leading to the release of polyunsaturated fatty acids (PUFAs), gamma-linolenic acid (GLA), arachidonic acid (AA), eicosapentaenoic acid (EPA), and docosahexaenoic acid (DHA), which form precursors to various pro- and anti-inflammatory eicosanoids. On the other hand, corticosteroids inhibit PLA2 activity and, thus, suppress the release of GLA, AA, EPA, and DHA. PUFAs and their metabolites have a negative regulatory action on the cGAS-STING pathway and, thus, suppress the inflammatory process and initiate inflammation resolution. Pro-inflammatory cytokines and corticosteroids (corticosteroids > IL-6, TNF-α) suppress desaturases, which results in decreased formation of GLA, AA, and other PUFAs from the dietary essential fatty acids (EFAs). A deficiency of GLA, AA, EPA, and DHA results in decreased production of anti-inflammatory eicosanoids and failure to suppress the cGAS-STING system. This results in the continuation of the inflammatory process. Thus, altered concentrations of PUFAs and their metabolites, and failure to suppress the cGAS-STING system at an appropriate time, leads to the onset of sepsis. Similar abnormalities are also seen in radiation-induced inflammation. These results imply that timely administration of GLA, AA, EPA, and DHA, in combination with corticosteroids and anti-IL-6 and anti-TNF-α antibodies, may be of benefit in mitigating radiation-induced damage and sepsis.

## 1. Introduction

Sepsis is the leading cause of death from infection. Its incidence is on the rise. In the United States alone, sepsis accounted for more than 20 billion dollars in hospital costs in 2011. Sepsis is defined as life-threatening organ dysfunction due to a dysregulated host response to infection. Septic shock is defined as a subset of sepsis, in which particularly profound circulatory, cellular, and metabolic abnormalities substantially increase mortality. The new diagnostic tool quickSOFA, or qSOFA, can be used to detect, at the bedside, patients who are at risk for sepsis. These are those who are experiencing:An alteration in mental statusA decrease in systolic blood pressure of less than 100 mm HgA respiration rate greater than 22 breaths/min

Studies suggest that patients with two or more of these conditions are at a significantly greater risk of having a prolonged ICU stay (3 or more days) or dying in the hospital. Hence, these patients need to be investigated further for organ dysfunction, therapy should be initiated or escalated, as appropriate, and the frequency of monitoring should be increased [1]. Despite these clinical indices in the evaluation of patients with sepsis, it will be worthwhile to have and/or establish specific laboratory indices that could serve as indicators of prognosis and response to therapy. Such laboratory indices could include: plasma cytokines, nitric oxide (NO), antioxidants (such as SOD, catalase, glutathione), lipid peroxides, ROS (reactive oxygen species), and pro- and anti-inflammatory eicosanoids. Several studies did look at these indices in sepsis and other critical illnesses, but a comprehensive study of these parameters was not done. In addition, there were no studies that looked at the potential interaction(s) among the various indices enumerated above with specific reference to sepsis. In the present review, some of these aspects are discussed.

Sepsis is characterized by life-threatening organ dysfunction caused by a dysregulated host response to infection, and it can occur after major surgery and injury. Dysregulated innate and adaptive immunity, as seen in those with sepsis, can result in sustained immunosuppression, predisposing them to secondary opportunistic infections. Paradoxically, autopsy studies revealed only minimal signs of inflammation or necrosis [2,3]. It is likely that the initial hyperinflammatory response(s) may result in the development of subsequent immunosuppression [4]. It is likely that the duration of the initial hyperinflammatory response and subsequent immunosuppression are variable. The heterogeneous presentation(s) and responses seen in sepsis may result in failure to recover from tissue injury, wound healing, and restoration of homeostasis. When dynamic changes in the innate and adaptive immune responses in sepsis was correlated with patient outcomes, it was found that absolute lymphocyte counts and lymphocyte subsets levels were reduced in those with sepsis, and there was an increase in the proportion of Tregs correlated with disease progression and immunosuppression, suggesting that downregulated adaptive immunity is responsible for the prolonged immune suppression seen in sepsis. In contrast, though cellular immunity reverted to near normal within 2 weeks of admission, humoral and innate immunity recovery lags. These findings suggest that appropriate therapeutic approaches need to be developed to improve the immune responses in those with sepsis.

This heterogenous presentation of sepsis could be attributed to mutations in Toll-like receptor-4 (TLR-4) [5,6,7,8], though this has been disputed [9]. Mutations in TLR-4 are associated with differences in lipopolysaccharide (LPS) responsiveness in humans, implying that host response to infections may be variable, rendering some to develop sepsis, while others may be resistant [5,6]. This indicates that an imbalance in the synthesis and release of pro-inflammatory and inflammation-resolving molecules determines the degree of infection, injury, and recovery from sepsis. Thus, a disparity in the timing and generation of adequate amounts of pro- and anti-inflammatory molecules determines the degree of the inflammatory process, inflammation resolution, and tissue repair in a sequential and coordinated fashion. TLRs regulate free radical generation, macrophage and leukocyte function, and modulate eicosanoid synthesis, and thus have a critical role in inflammation, immune response, and development and/or recovery from sepsis [10,11,12]. Hence, efforts need to be made to revert the initial hyperinflammatory response and subsequent immunosuppression to facilitate recovery from sepsis to one in which there could be a role for essential fatty acids (EFAs) and their metabolites. This is further supported by the observation that EFAs and their metabolites have a regulatory role in the elaboration of various cytokines, reactive oxygen species (ROS), TLR and NF-kB expression, and the cGAS-STING pathway (see below).

## 2. TLRs and Eicosanoids in Sepsis

Polyunsaturated fatty acids (PUFAs) form an important constituent of all cell membranes and regulate cell membrane fluidity and the expression of receptors on their surface and serve as mechanotransducers to convey external stimuli to the cytoplasm and DNA [13]. This cross-communication between the cell membrane PUFAs and the nucleus (and, consequently, genes) enables the cell to tailor its responses to external stimuli and produce adequate changes in cell shape and control dynamic cell behavior [13,14,15,16]. Thus, cells sense their physical environment through their plasma membrane, which harbors mechanosensitive ion channels and adhesion molecules. It is likely that cells sense their environment though the cell membrane that induces stretch in the nuclear membrane, leading to the activation of the enzyme cytosolic phospholipase A2 (cPLA2), which induces the release of cell membrane PUFAs (especially arachidonic acid, AA). The released AA, in turn, initiates cell blebbing and movements, as well as other cell functions [13]. It is also known that cPLA2 senses nuclear swelling upon osmotic shock to initiate rapid immune cell chemotaxis [17,18]. AA can be converted to form various eicosanoids, which attract immune cells and control cell differentiation and survival, among various other functions (see Figure 1 for metabolism of essential fatty acids). These studies suggest that the fluidity of plasma and nuclear membranes sense physical clues from the cell environment and converts them into chemical signals that drive inflammation. It is noteworthy that cPLA(2)(−/−) (cPLA2 knockout) mice recover from allergen-induced bronchoconstriction and show no airway hyperresponsiveness. Their peritoneal macrophages [cPLA(2)(−/− mice] do not produce prostaglandins (PGs), leukotriene B4 (LTB4), and cysteinyl leukotrienes after stimulation. Moreover, cPLA(2)(−/−) mice bone marrow-derived mast cells also do not produce eicosanoids. Thus, cPLA2 has a critical role in inflammation and tissue injury that is relevant to the pathobiology of sepsis.

PUFAs modulate the expression of TLRs. COX-2 (cyclooxygenase-2) mediated high production of PGE2 following LPS stimulation, whereas LPS down-regulated COX-1, and COX-1 deficiency enhanced PGE2 production. Thus, coordinated down-regulation of COX-1 facilitates PGE2 production after TLR activation [19,20,21]. On the other hand, the supplementation of AA and docosahexaenoic acid (DHA) inhibited intestinal TLR-4 gene expression and ameliorated NEC (necrotizing enterocolitis) [22,23]. Resolvins and protectin D1, which are derived from DHA, inhibited the number of infiltrating leukocytes, blocked TLR-mediated activation of macrophages, and suppressed ischemia-reperfusion-induced kidney injury [24,25]. These results emphasize the fact that the coordinated synthesis, release, and action of pro- and anti-inflammatory eicosanoids can control inflammation and its resolution, including in sepsis [23,24,25,26,27,28].

## 3. The cGAS-STING Pathway, MN Cells, and Sepsis

Microbial infection (bacteria, viruses, and fungal), the most common cause of sepsis, release DNA into cells via endocytosis or phagocytosis, or release DNA directly into the infected cells and/or macrophages. Foreign DNA in the host cell is recognized and eliminated by various nucleases. Failure of this process leads to the triggering of inflammation by the production of type 1 interferons (IFNs), tumor necrosis factor-α (TNF-α), and IL-18. In addition, extensive host cell damage due to infection(s) and the release of their DNA could be a major source of released DNA during sepsis [29]. Circulating cell-free DNA (cfDNA and mitochondrial DNA: mtDNA), released by host cells that are significantly elevated in those with sepsis, can cause inflammation and organ failure [30,31,32]. Hence, circulating mtDNA (mitochondrial DNA) can be used as a marker of severity and prognostic factor in sepsis, since mtDNA initiates inflammation and subsequent immunosuppression. The concentration of cfDNA (circulating free DNA) in blood varies significantly; it ranges between 0–5 and >1000 ng/mL in patients with cancer, and between 0 and 100 ng/mL in healthy subjects. There is also a marked variation in blood ctDNA levels among patients with different tumor types and diseases. Elevated circulating cfDNA could be used as a marker of severity of sepsis [33]. In addition, circulating mtDNA can activate Toll-like receptors (especially TLR9) and inflammasomes, contributing to sepsis [34,35].

Cyclic GMP-AMP (cGAMP) synthase (cGAS) present in the cytosol of cells acts as a DNA sensor. It detects foreign DNA and mounts an immune response. The binding of DNA to cGAS produces the cyclic dinucleotide second messenger 2′3′-cGAMP, which activates the stimulator of interferon genes (STING), resulting in the release of interferons (IFNs). Inhibiting cGAS activity suppresses IFN-stimulated gene expression and ameliorates autoimmune diseases and sepsis [28,32,36,37,38,39,40,41,42]. Thus, the CGAS-STING pathway is not only important in host defense, but has a critical role in sepsis and autoimmune diseases.

At the bedside, the cytoplasmic DNA can be recognized as micronuclei (MN) in the circulating lymphocytes, buccal mucosa, and bone marrow cells (see Figure 2). The quantification of the number of MN-containing cells can be done easily and can be used as a measure of severity of the underlying disease [28,43,44,45,46]. Since intracellular DNA triggers the inflammatory process and, consequently, the development of sepsis and autoimmune diseases, intracellular DNA/RNA should be cleared effectively in a timely fashion. This is done by cellular endonucleases (DNase I, DNase II, and DNase III), suggesting that their deficiency can result in the development of severe sepsis. These results emphasize that the DNA sensor cGAS-STING pathway triggers an inflammatory process seen in several inflammatory conditions including sepsis, cancer, and autoimmune diseases [28,33,41]. Experimental animals deficient in STING have impaired immuno-surveillance [33], indicating that adequate inflammation is needed to initiate anti-inflammatory pathways to restore homeostasis. In sepsis, it is likely that the activated cGAS-STING pathway results in pro-inflammatory gene expression, but is unable to trigger the much-needed anti-inflammatory pathways to restore tissue homeostasis. Hence, understanding the endogenous mechanisms/molecules that regulate the cGAS-STING pathway and developing methods of restoring the imbalanced pro- and anti-inflammatory pathways is needed to induce recovery from sepsis. In this context, PUFAs and their metabolites seem to have a critical role in the regulation of the cGAS-STING pathway, in regards to inflammation and its resolution.

## 4. Metabolism of Essential Fatty Acids (EFAs)

Dietary cis-linoleic acid (LA, 18:2 n-6) and alpha-linolenic acid (ALA, 18:3 n-3) are essential fatty acids (EFAs) that are converted to their long-chain metabolites by the action of Δ^6^ and Δ^5^ desaturases. Thus, desaturases are the rate-limiting enzymes in the metabolism of EFAs. LA is converted to form gamma-linolenic acid (GLA, 18:3 n-6) by the action of Δ^6^ desaturase. GLA is converted to dihomo-gamma-linolenic acid (DGLA, 20:3 n-6) by an elongase. DGLA, in turn, is converted to arachidonic acid (AA, 20:4 n-6) by the action of Δ^5^ desaturase. Similarly, ALA is converted to eicosapentaenoic acid (EPA, 20:5 n-3) and docosahexaenoic acid (DHA, 22:6 n-3) by the action of these desaturases and elongases. DGLA is the precursor of 1 series of prostaglandins (PGs), whereas AA is the precursor of 2 series PGs, prostacyclin (PGI2), and thromboxanes (TXs), and 4 series leukotrienes (LTs). On the other hand, EPA is the precursor of 3 series PGs and TXs and 5 series LTs. Most of these PGs, LTs, and TXs are pro-inflammatory in nature. AA is also the precursor of lipoxin A4 LXA4), a potent anti-inflammatory molecule. EPA gives rise to anti-inflammatory E resolvins, whereas DHA is the precursor of anti-inflammatory resolvins of D series, protectins, and maresins (see Figure 1). PGE1, PGI2, and PGJ2 (also obtained from AA) have anti-inflammatory actions. Thus, AA and EPA are the precursors of both pro- and anti-inflammatory eicosanoids, whereas DHA is the precursor of only anti-inflammatory resolvins, protectins, and maresins. M1 and M2 macrophages and T_H_1 and T_H_2 cells bring about their pro- and anti-inflammatory actions, respectively, by elaborating pro- and anti-inflammatory eicosanoids. Thus, AA, EPA, and DHA not only modify cell membrane fluidity, but also regulate inflammation and immune response [13,28,41,47,48,49,50,51].

LA, GLA, DGLA, AA, ALA, EPA, DHA, PGE1, PGE2, PGE3, PGI2, lipoxins, resolvins, protectins, and maresins suppress the production of pro-inflammatory cytokines, especially IL-6, TNF-α, and HMGB1, and enhance the production of anti-inflammatory IL-4 and IL-10 cytokines and vice versa. IL-6, TNF-α, and HMGB1 augment the production of PGE2 and LTs. IL-6 and TNF-α suppress the activities of desaturases and, thus, induce deficiency of GLA, DGLA, and AA, and EPA and DHA. This feedback regulation between cytokines, as well as EFA metabolism (see Figure 3), implies that maintaining adequate cell/tissue concentrations of GLA, DGLA, AA, EPA, and DHA is needed to suppress inappropriate and excess generation of IL-6, TNF-α, and that HMGB1 is harmful, especially in sepsis. In addition, IL-6 and TNF-α enhance the production of PGE2, especially by macrophages, whereas PGE2 is a potent suppressor of IL-6 and TNF-α secretion. These interaction(s) among EFAs and their metabolites, cytokines, and desaturases need to be considered whenever the actions of these molecules are considered in sepsis and other inflammatory conditions. In addition, it is noteworthy that PGs, TXs, and LTs enhance the expression of NF-κB, whereas AA, EPA, and DHA, lipoxins, resolvins, protectins, and maresins suppress its (NF-κB) expression [13,28,41,47,48,49,50,51] (see Figure 3). Thus, restoring the balance among various EFAs and their metabolites, cytokines, desaturases, COX, and LOX enzymes is needed to suppress inappropriate inflammatory events in sepsis, in order to restore homeostasis (see Figure 3). In addition, PUFAs have been shown to suppress inappropriate activation of the cGAS-STING pathway, emphasizing the role of EFAs and their metabolites in sepsis and other inflammatory conditions (see Figure 3)

## 5. The cGAS-STING System, MN Cells, and EFAs and their Metabolites in Inflammation and Sepsis

Previously, we reported that γ-radiation, benzo(a)pyrene, 4-α-phorbol, and diphenylhydantoin (DPH), an anti-epileptic drug, induced DNA damage and the formation of an increased number of MN-containing bone marrow cells in experimental animals and human peripheral lymphocytes, which can be suppressed by GLA, DGLA, AA, PGE1, PGE2, TXB2, and PGI2 [52,53,54,55,56]. Oral GLA supplementation to patients of epilepsy on long-term DPH therapy decreased the number of MN-containing peripheral lymphocytes [56,57], which we attributed to increased formation and accumulation of toxic lipid peroxides in the MN-containing cells and, consequently, their apoptosis. This is supported by the results of our studies, in which we observed that virus-infected and tumorous cells undergo apoptosis when supplemented with GLA, AA, EPA, and DHA, correlated to the increased formation and accumulation of toxic lipid peroxides, which are only in the tumor, not in normal cells [47,58,59,60,61,62,63]. These results suggest that MN-containing cells have a defective antioxidant defense system and so accumulate toxic lipid peroxides; hence, they undergo apoptosis. On the other hand, normal cells are able to enhance their antioxidant defenses (especially glutathione levels) such that they are able to nullify the cytotoxic actions of lipid peroxides. In fact, we found that, once the accumulation of toxic lipid peroxides reached a critical level, the normal cells were able to degrade the lipid peroxides and eliminate them from the cells; thus, they survived. These results emphasize the distinct differences between the normal and tumor cells in their response to accumulation of toxic lipid peroxides [58,59,60]. In this context, it is noteworthy that tumor cells that are resistant to the actions of radiation and chemotherapeutic drugs have enhanced GPX4 (glutathione peroxidase) activity, implying that they are able to withstand the toxic effects of lipid peroxides due to their enhanced antioxidant capacity. These results suggest that tumoricidal action of radiation and chemotherapeutic drugs can be augmented by making efforts that enhance the lipid peroxidation process and result in the accumulation of lipid peroxides in tumor cells [61,62,63,64]. These results imply that the balance between antioxidants and the lipid peroxidation process is vital to cell survival. In this context, the recognition of cytoplasmic nuclear material by the cell (and, consequently, the release of pro-inflammatory cytokines), their ability to induce the formation and release of free radicals (including ROS-reactive oxygen species), and the regulation of these processes by lipids (especially EFs and their metabolites) assumes significance.

The presence of nuclear material in the cytoplasm (in the form of micronuclei) results in the activation of the cGAS-STING system, which leads to increased expression of NF-κB. This leads to increased production of IL-6, TNF-α, and IFNs, and free radicals, events that result in the formation and accumulation of toxic lipid peroxides in MN (nuclear material)-containing cells, which induces their apoptosis. In addition, these MN-containing abnormal cells are recognized by immunocytes and induce their apoptosis, a process in which GLA and other fatty acids have a role [47,58,59,60,61,62,63]. We observed that GLA, AA, LXA4, resolvins, and protectins suppress NF-κB and COX-2 expression and decrease plasma and tissue levels of IL-6 and TNF-α, emphasizing their anti-inflammatory action [65,66,67,68,69].

MN-containing cells trigger inflammation by activating the cGAS-STING system, which can occur even in sepsis, and other inflammatory conditions, including, but not limited to, radiation-induced tissue damage [70,71,72,73,74,75,76,77]. I propose that the degree of inflammation in sepsis and other inflammatory conditions can be assessed by counting the number of MN-containing circulating lymphocytes, buccal mucosal, and bone marrow cells [71,72,73].

## 6. Radiation-Induced DNA Damage, Activation of the cGAS-STING System, and EFAs

Radiation is a potent DNA-damaging agent and stimulator of the cGAS-STING pathway that accounts for its various inflammatory, immunological, and anti-cancer actions [74,75,76,77]. Radiation activates PLA2, COX-2, and LOX enzymes, which leads to the release of GLA, DGLA, AA, EPA, and DHA from the cell membrane lipid pool and their subsequent conversion to various pro- and anti-inflammatory eicosanoids (PGs, LTs, TXs, and lipoxins, resolvins, protectins, and maresins) that modulate various actions of radiation (see Figure 3). It is noteworthy that IL-6, TNF-α, and IFNs are potent stimulators of PLA2. The fatty acids released because of PLA2 activation get converted, not only to form pro-inflammatory PGs, LTs, and TXs, but also to anti-inflammatory LXA4, resolvins, protectins, and maresins. Several of these eicosanoids (especially PGE1, PGE2, PGI2, lipoxins, resolvins, protectins, and maresins) suppress the cGAS-STING system, and NF-κB, IL-6, and TNF-α expression, suggesting a negative and positive feedback regulation among radiation, cytokines, lipids, and inflammation. GLA, DGLA, PGE1, lipoxins, resolvins, protectins, and maresins not only suppress pro-inflammatory cytokine (IL-6, TNF-α, HMGB1) production but also protect against radiation-induced genetic damage, suggesting that lipids may serve as endogenous negative regulators of the cGAS-STING system (see Figure 3). This is supported by the recent report that STING regulates PUFAs metabolism, and, in turn, PUFAs inhibit STING-dependent inflammation—a cross-regulation that seems to be central to the maintenance of metabolic homeostasis [78,79]. The STING (stimulator of interferon genes) protein inhibits the fatty acid Δ^6^ and Δ^5^ desaturases, the rate-limiting enzymes in the conversion of dietary EFAs, LA, and ALA, to their respective long-chain metabolites. STING ablation enhanced the activity of desaturases and led to accumulation of PUFAs that inhibit the cGAS-STING system. On the other hand, PUFAs inhibited STING, which led to the regulation of antiviral responses and contributed to resolving STING-associated inflammation. Thus, a negative regulatory feedback loop seems to exist between STING and desaturases to fine tune inflammatory responses. These results imply that administration of GLA/DGLA/AA/ EPA/DHA can suppress the cGAS-STING pathway and NF-kB, IL-6, and TNF-α expressions and, thus, suppress inappropriate inflammatory responses seen in sepsis. These results suggest that deficiency of GLA/DGLA/AA/EPA/DHA can result in enhanced expression of the cGAS-STING pathway, NF-κB, and increased generation of IL-6 and TNF-α (see Figure 3). These results are in support of the previous proposal [28]: that EFAs and their metabolites have a regulatory role in the expression and function of the cGAS-STING pathway and, thus, regulate inflammation, immune response, and resolution of inflammation. GLA/AA/EPA/DHA released due to the activation of PLA2 by IL-6, TNF-α, and IFNs are converted to both pro- and anti-inflammatory metabolites that have both positive and negative feedback control on inflammation and its resolution [28]. These results are supported by the observation that, in those with sepsis, there is a deficiency of GLA, DGLA, AA, and EPA ([80], see Table 1). These results imply that infusion or supplementation of the deficient fatty acids suppress the expression of the cGAS-STING system, NF-κB, IL-6, and TNF-α, and, thus, inhibit inappropriate inflammatory and immune responses [28]. This is supported by the results of our study with radiation, wherein we observed that lethal radiation-indued mortality (similar to the high mortality seen in sepsis) can be reduced from 80% to 20% by the administration of GLA. The dynamic changes in the plasma levels of IL-6, TNF-α, IL-10, and HMGB1 vs expressions of desaturases/COX-2/LOX enzymes vs PGE2/LTE4/LXA4 levels that are seen because of GLA administration is rather interesting [80], see Figure 4, Figure 5 and Figure 6. In this study, radiation-induced alterations in the plasma levels of pro- and anti-inflammatory cytokines and eicosanoids, and expressions of desaturases and COX and LOX enzymes were restored to near normal by GLA treatment. The increased number of MN cells induced by radiation were also restored to near normal by GLA administration. Thus, radiation-induced DNA damage (in the form of an increased number of MN cells), which activates the cGAS-STING system and, consequently, results in changes in cytokines and eicosanoids and alterations in the expression of desaturases and COX and LOX enzymes, can be restored to normal by GLA, likely due to its (GLA) deficiency induced by radiation (see Figure 3). Since both radiation and sepsis are severe inflammatory conditions that have several overlapping features, and as sepsis is associated with a deficiency of GLA/DGLA/AA/EPA, it is reasonable to understand why GLA administration is beneficial. In this context, it is noteworthy that radiation and several viruses suppress the expression of desaturases and activate COX-2 and LOX enzymes that lead to the development of a deficiency of GLA/DGLA/AA/EPA/DHA and the increased formation and action of pro-inflammatory cytokines and eicosanoids. It is likely that, in the presence of AA/EPA/DHA deficiency, production of pro-inflammatory PGE2 and LTs are increased, while that of LXA4/resolvins/protectins/maresins are decreased. Meanwhile, supplementation of GLA/DGLA/AA/EPA/DHA enhances the production of LXA4/resolvins/protectins/maresins that induce resolution of inflammation [28,47,49,50,69,81,82,83].

## 7. Conclusions and Therapeutic Implications

It is evident from the preceding discussion that sepsis and other inflammatory conditions, such as radiation-induced tissue damage, cause significant DNA damage, activating the cGAS-STING pathway and leading to an increase in the expression of NF-κB, TNF-α, IL-6, and other pro-inflammatory cytokines. Concomitantly, there is an increase in the expression of COX-2 and LOX enzymes. All these events lead to the initiation of sepsis. Activation of the cGAS-STING pathway seen in sepsis and radiation suppresses the expression of desaturases, resulting in decreased formation of GLA/DGLA/AA/EPA/DHA. Deficiency of these PUFAs results in the accumulation of MN cells, the activation of the cGAS-STING system and NF-κB, and the production of increased amounts of IL-6, TNF-α, and HMGB1 cytokines. These events produce intense inflammation, seen in sepsis and radiation injury. This is supported by the observation that patients with septicemia have low plasma concentrations of GLA/DGLA/AA/EPA ([80], see Table 1). The observation that GLA is beneficial against radiation-induced DNA damage and improved survival lends support to this proposal [80]. Hence, it is important to measure, not only plasma or tissue concentrations of various cytokines in sepsis, but also the expression of the cGAS-STING pathway, NF-κB, desaturases, COX and LOX enzymes, and various eicosanoids and cytokines to know the inflammatory and anti-inflammatory balance in sepsis and radiation-induced damage, and in other inflammatory conditions.

This concept also explains why the administration of corticosteroids is of little benefit in sepsis. Previously, we and others showed that corticosteroids suppress the expression of desaturases in addition to their ability to inhibit PLA2, COX-2, and LOX enzymes, which results in decreased formation of various eicosanoids. Corticosteroids, by virtue of their ability to suppress the expression of desaturases, induce a deficiency of GLA/DGLA/AA/EPA/DHA, which are needed for the formation of PGs/LTS/ TXs and LXA4/resolvins/protectins/maresins, essential for inducing much-needed inflammation and the subsequent initiation of the inflammation resolution process and to facilitate wound healing. Thus, corticosteroids, by inducing decreased formation of GLA/DHLA/AA/EPA/DHA from LA and ALA, and the resultant deficiency of LXA4, resolvins, protectins, and maresins, are of little benefit in sepsis and other inflammatory conditions [27,28,84,85,86]. This implies that co-administration of GLA/DHLA/AA/EPA/DHA, along with corticosteroids and other co-factors needed for synthesis of anti-inflammatory eicosanoids such as vitamin C, B, B6, B12, and folic acid, will be of significant benefit in the amelioration of sepsis.

In view of the arguments and evidence presented here, it is suggested that a combination of corticosteroids and/or anti-IL-6 and anti-TNF-α antibodies and GLA, DHLA, AA, EPA, DHA, along with vitamins C, B1, B6, B12, and folic acid, can prevent and suppress sepsis and other inflammatory conditions.

## Figures and Tables

**Figure 1 biomolecules-13-01332-f001:**
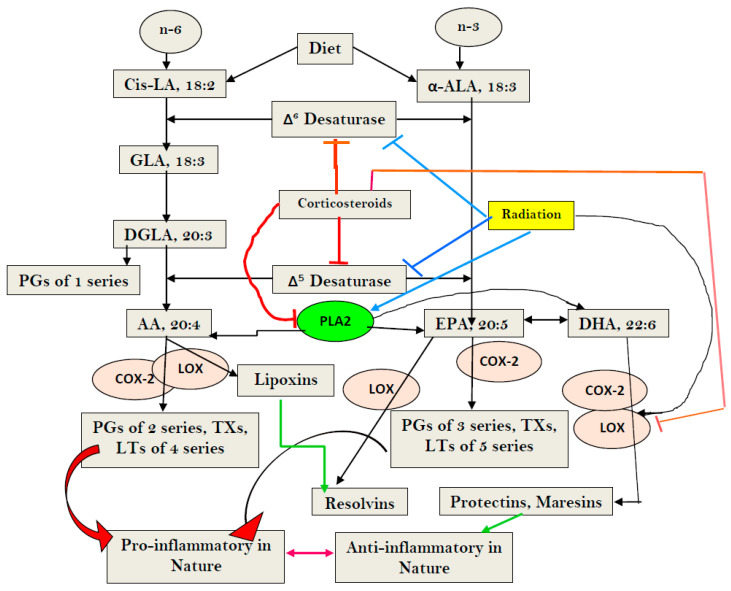
Scheme showing metabolism of essential fatty acids (EFAs) and the effect of corticosteroids on the same. Note the differences in the actions of corticosteroids (dexamethasone) and radiation. Corticosteroids inhibit desaturases, and PLA2, COX-2, and LOX enzymes, whereas radiation blocks only desaturases, but activates PLA2, COX-2, and LOX enzymes. Radiation enhances the formation of PGs, LTs, and TXs, but blocks the formation of LXA4, resolvins, protectins, and maresins by inducing deficiency of AA/EPA/DHA, whereas corticosteroids block the formation of PGS, LTs, TXs, and LXA4, resolvins, protectins, and maresins by inhibiting PLA2, COX-2, and LOX enzymes, and induces deficiency of AA/EPA/DHA by inhibiting desaturases. Thus, corticosteroids are potent inhibitors of inflammation but also block resolution of inflammation, whereas radiation is a potent inducer of inflammation and interferes with the inflammation resolution process.

**Figure 2 biomolecules-13-01332-f002:**
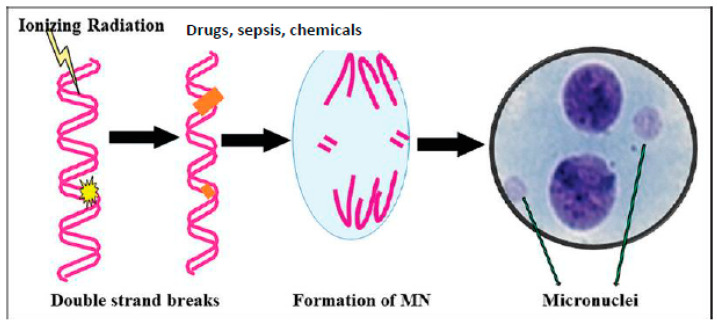
Scheme of formation of MN cells. All agents that damage DNA can cause the formation of MN cells. Microbes, when they enter the cell, shed their DNA/RNA; such cytoplasmic nuclear material is recognized by the cGAS-STING system, leading to an inflammatory response, as evidenced by enhanced levels of IL-6, TNF-α, HMGB1, IFNs, and ROS, and the increased formation of lipid peroxides, both in the plasma and in the cells (see also Figure 3).

**Figure 3 biomolecules-13-01332-f003:**
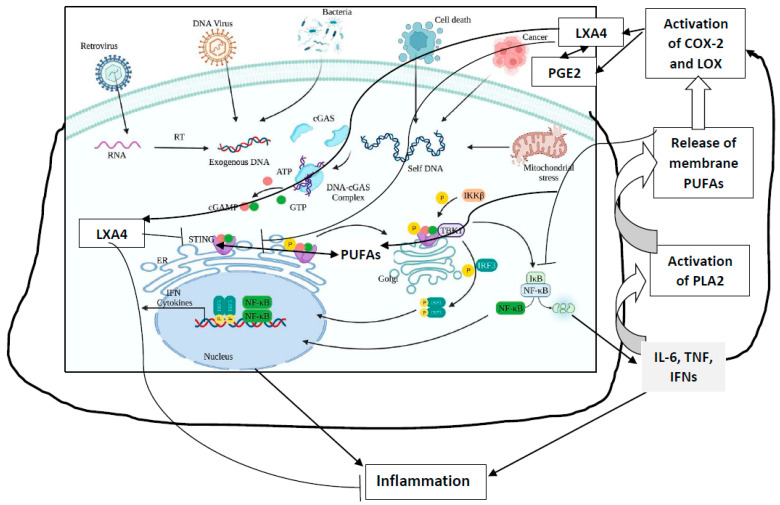
Scheme showing formation of MN cells and activation of the cGAS-STING pathway and NF-κB. Cytokines activate the PLA2 enzyme, resulting in the release of PUFAs from the cell membrane. PUFAs (especially AA) are converted to PGE2, LXA4, and other respective metabolites that modulate the production of IL-6, TNF, and IFN. PUFAs, PGE2, and LXA4 suppress NF-kB activation and, thus, suppress the inflammatory process. LXA4 and PGE2 suppress the cGAS-STING pathway. IL-6, TNF, and IFN-γ augment PGE2 formation, which has both pro- and anti-inflammatory actions. Once the inflammation process reaches its threshold, PGE2 triggers the production of LXA4, which initiates inflammation resolution. Increased formation of LXA4 is due to the activation of 5- and 12-LOX enzymes induced by PGE2, which redirects AA to generate LXA4. The cGAS-STING pathway is suppressed by LXA4/AA/EPA/DHA. Thus, the availability of adequate amounts of AA/EPA/DHA and the formation of LXA4, resolvins, protectins, and maresins suppress the inflammatory process.

**Figure 4 biomolecules-13-01332-f004:**
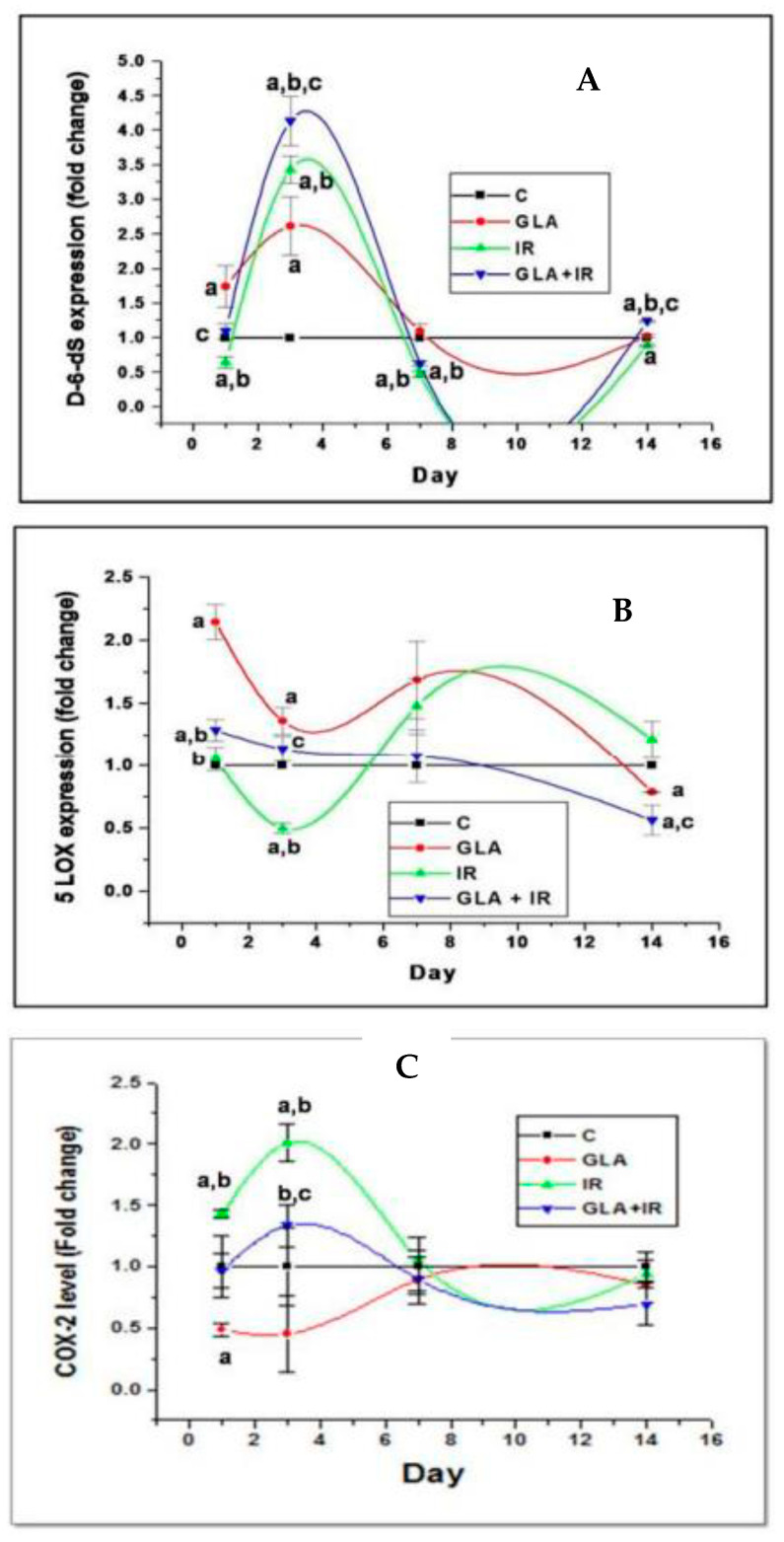
Effect of whole-body lethal radiation and GLA treatment on the expressions of delta-6-desaturase (**A**), COX-2 (**C**), and 5-LOX (**B**) enzymes. This data is taken from reference no. [80]. Effect of GLA and irradiation (7.5 Gy) on genes associated with PUFA metabolism in duodenum tissue on day1, day3, day 7 and day14. Values (n = 3) expressed as mean ± SEM. ^a,b,c^
*p* < 0.01 when compared with control, GLA and irradiation.

**Figure 5 biomolecules-13-01332-f005:**
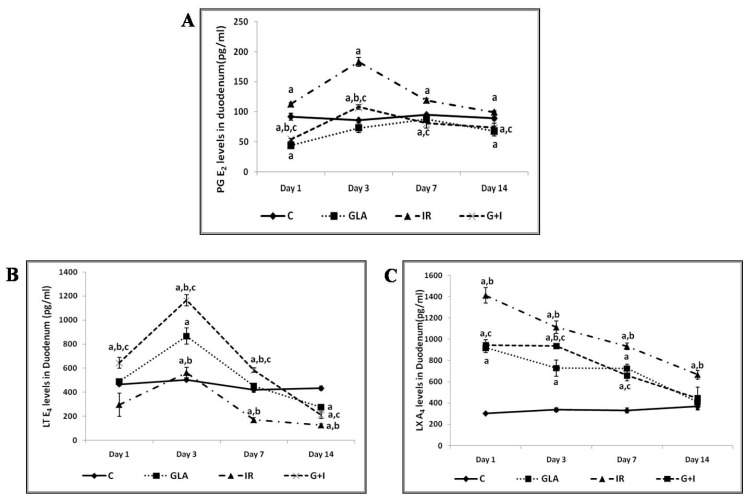
Effect of radiation and GLA on plasma levels of PGE2 (**A**), LTE4 (**B**), and LXA4 (**C**). Note the dynamic equilibrium among these eicosanoids from time to time. Estimation of PUFA metabolites PG E_2_ (**A**), LT E_4_ (**B**) and LXA_4_ (**C**) levels by ELISA method. Mouse were pretreated with 100 μg/kg GLA at 48, 24 and 1 h prior to irradiation and were subjected to total body irradiation of 7.5 Gy (at 1 Gy/min) and duodenum samples were obtained on day 1, day 3, day 7 and day 14 post-irradiation. All the values are expressed as mean ± SEM (n = 3). Statistical significance was calculated using *t*-test and *p*-value (<0.001) represented as ^a,b,c^ when compared to control, GLA and IR alone respectively. C- Control, GLA- Gamma-linolenic acid, IR- Irradiation (7.5 Gy).

**Figure 6 biomolecules-13-01332-f006:**
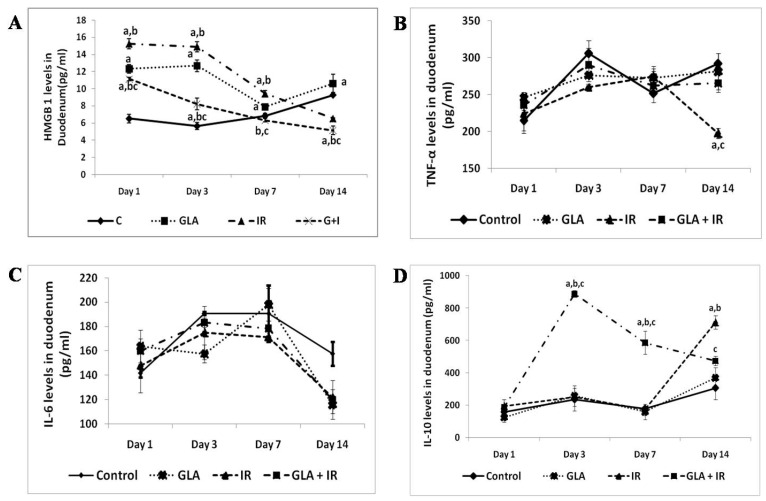
Effect of radiation and GLA on plasma levels of HMGB1 (**A**), TNF (**B**), IL-6 (**C**), and IL-10 (**D**). Estimation of cytokines HMGB1 (**A**),TNF-α (**B**), IL-6 (**C**) and IL-10 (**D**) levels by ELISA method. Mouse were pretreated with 100 μg/kg GLA at 48, 24 and 1 h prior to irradiation and were subjected to total body irradiation of 7.5 Gy (at 1 Gy/min) and duodenum samples were obtained on day 1, day 3, day 7 and day 14 post-irradiation. All the values are expressed as mean ± SEM (n = 3). Statistical significance was calculated using *t*-test and *p*-value (<0.05) represented as ^a,b,c^ when compared to control, GLA and IR alone respectively. This data is from reference no. [80].

**Table 1 biomolecules-13-01332-t001:** Fatty acid analysis of plasma phospholipid fraction in patients with septicemia.

Fatty Acid	Control (n = 10)	Septicemia (n = 14)
16:0	24.8 ± 3.4	26.95 ± 1.1
18:0	23.3 ± 4.1	24.58 ± 6.0
18:1 n-9	13.1 ± 2.3	16.5 ± 3.3 *
18:2 n-6 (LA)	17.7 ± 3.1	16.3 ± 3.3
18:3 n-6 (GLA)	0.13 ± 0.09	0.04 ± 0.05 *
20:3 n-6 (DGLA)	3.2 ± 0.79	0.46 ± 0.54 *
20:4 n-6 (AA)	8.8 ± 2.0	5.8 ± 1.6 *
18:3 n-3 (ALA)	0.27 ± 0.12	0.16 ± 0.11 *
20:5 n-3 (EPA)	0.25 ± 0.26	0.01 ± 0.01 *
22:6 n-3 (DHA)	1.43 ± 0.43	1.2 ± 1.14

All values are expressed as mean ± S. E. * *p* < 0.05 compared to control. This data is taken from reference no. [79].

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
