# Peer review of "Infection, Inflammation, and Immunity in Sepsis"

_biomolecules, 2023, doi:10.3390/biom13091332_

Round 1

Reviewer 1 Report (Previous Reviewer 2)

The revised manuscript is improved and the author's responses are satisfactory.

Minor editing of English language required

Reviewer 2 Report (Previous Reviewer 1)

The revised manuscript has quite impressive effort to make the article more reachable and interesting to the readers. Also the addibg meaningful mechanistic images gave this to reflect importance of this topic. Congratulations to the authors.

This manuscript is a resubmission of an earlier submission. The following is a list of the peer review reports and author responses from that submission.

Round 1

Reviewer 1 Report

Working in the area of sepsis, i dont feel this review article contributes any new approach and new informations to the readers. The subtopic or points selected in this review are almost similar to the other reviews on sepsis. Insteaf of writing on same subtopics in sepsis, it is suggested to the author to re-write the review article with new subarea or subtopic instead of using same focus which already so many other reviews on sepsis have been reported and discussed. 

Ok. Could be better for an international journal publication level.

Author Response

I am resubmitting the review on sepsis that has been modified as per the suggestions of the reviewers and editor's note.

The manuscript has been edited and modified as per the suggestions. Some of the subheadings, text and references have been modified.

I may add here that some of the references referring to our previous work are indeed necessary as we are the first to show that MN cells incidence can be modified by PUFAS back in 1980's. WE are also the first to propose and show that PUFAs can have a role in sepsis and cGAS-STING system can be modulated by PUFAs. I have also added new references in the modified manuscript.     

Reviewer 2 Report

In this review, " Infection, Inflammation, and Immunity in Sepsis”  The author, Undurti N. Das,  

 based on published data, has suggested that a combination of corticosteroids and/or anti-IL-6 and anti-TNF-α antibodies and GLA.DGLA, AA, EPA, DHA is needed to prevent and manage sepsis and other inflammatory conditions.

The comments and suggestions for this manuscript are as follows-

1.     The introduction and the main body of the manuscript are typical textbook types. This is lacking intellectual input from authors. The author should provide a comprehensive introduction and discussion.

2.     Page 4, lines 150-153. The author has stated that “At the bedside, the cytoplasmic DNA can be recognized as micronuclei (MN) in the circulating lymphocytes, buccal mucosa, and bone marrow cells. The quantification of the number of MN-containing cells can be done easily and used as a measure of severity of the underlying disease”. My query is what is the amount (ng or pg) of CF DNA considered as the threshold amount for the diagnostic purpose? Consider that there will be always free DNA due to natural immune cells' death.

3.     Figure 4, 5Lox image is super saturated, which may give false results. However, since this result is already published (ref 81), it can be removed from the main manuscript. This is not adding any new information.

4.     Around 33% of citations (DNS, either first or co-authors) for this manuscript are self-cited. The author must remove inappropriate references.

Minor editing is required.

Author Response

Comment: The introduction and the main body of the manuscript are typical textbook types. This is lacking intellectual input from authors. The author should provide a comprehensive introduction and discussion.

Response: The Introduction has been modified as suggested.

Comment:

  1. Page 4, lines 150-153. The author has stated that “At the bedside, the cytoplasmic DNA can be recognized as micronuclei (MN) in the circulating lymphocytes, buccal mucosa, and bone marrow cells. The quantification of the number of MN-containing cells can be done easily and used as a measure of severity of the underlying disease”. My query is what is the amount (ng or pg) of CF DNA considered as the threshold amount for the diagnostic purpose? Consider that there will be always free DNA due to natural immune cells' death.
  2. Response: As suggested by the reviewer new information about  the circulating free DNA has been added. THe suggestion that the number of circulating MN can be used to estimate the severity of an inflammatory diseases is a new one.  
  3. Comment:
  4. Figure 4, 5Lox image is super saturated, which may give false results. However, since this result is already published (ref 81), it can be removed from the main manuscript. This is not adding any new information.
  5. Response: The figure 4 has been modified as suggested and only the relevant data is provided.
  6. comment: 
  7. Around 33% of citations (DNS, either first or co-authors) for this manuscript are self-cited. The author must remove inappropriate references.
  8. Response: I feel that all the references quoted are appropriate. For instance the references pertaining to MN and PUFAs-we are the first to show that the number of MN cells can be reduced by GLA and PGE1 and PGI2 and hence these references are quoted. Similarly we are the first to suggest that there is a role for PUFAs in sepsis is also mentioned first by us and so there references are quoted. WE are also the frist to suggest that cGAS-STING system can be modified by PUFAs that has now been confirmed. In addition, a couple of new references have been added in the modified version.